# Antibacterial Activity and Mechanisms of TroHepc2-22, a Derived Peptide of Hepcidin2 from Golden Pompano (*Trachinotus ovatus*)

**DOI:** 10.3390/ijms24119251

**Published:** 2023-05-25

**Authors:** Zhengshi Zhang, Yongcan Zhou, Han Zhang, Xiangyu Du, Zhenjie Cao, Ying Wu, Chunsheng Liu, Yun Sun

**Affiliations:** 1Sanya Nanfan Research Institute, Hainan University, Sanya 572022, China; 2Collaborative Innovation Center of Marine Science and Technology, Hainan University, Haikou 570228, China

**Keywords:** *Trachinotus ovatus*, hepcidin, antimicrobial peptide, TroHepc2-22, antibacterial mechanisms

## Abstract

Hepcidin, a cysteine-rich antimicrobial peptide, has a highly conserved gene structure in teleosts, and it plays an essential role in host immune response against various pathogenic bacteria. Nonetheless, few studies on the antibacterial mechanism of hepcidin in golden pompano (*Trachinotus ovatus*) have been reported. In this study, we synthesized a derived peptide, TroHepc2-22, from the mature peptide of *T. ovatus* hepcidin2. Our results showed that TroHepc2-22 has superior antibacterial abilities against both Gram-negative (*Vibrio harveyi* and *Edwardsiella piscicida*) and Gram-positive (*Staphylococcus aureus* and *Streptococcus agalactiae*) bacteria. Based on the results of a bacterial membrane depolarization assay and propidium iodide (PI) staining assay in vitro, TroHepc2-22 displayed antimicrobial activity by inducing the bacterial membrane depolarization and changing the bacterial membrane permeability. Scanning electron microscopy (SEM) visualization illustrated that TroHepc2-22 brought about membrane rupturing and the leakage of the cytoplasm for the bacteria. In addition, TroHepc2-22 was verified to have hydrolytic activity on bacterial genomic DNA in view of the results of the gel retardation assay. In terms of the in vivo assay, the bacterial loads of *V. harveyi* in the tested immune tissues (liver, spleen, and head kidney) were significantly reduced in *T. ovatus*, revealing that TroHepc2-22 significantly enhanced the resistance against *V. harveyi* infection. Furthermore, the expressions of immune-related genes, including tumor necrosis factor-α (*TNF-α*), interferon-γ (*IFN-γ*), interleukin 1-β (*IL-1β*), *IL-6*, Toll-like receptor 1 (*TLR1*), and myeloid differentiation factor 88 (*MyD88*) were significantly increased, indicating that TroHepc2-22 might regulate inflammatory cytokines and activate immune-related signaling pathways. To summarize, TroHepc2-22 possesses appreciable antimicrobial activity and plays a vital role in resisting bacterial infection. The observation of our present study unveils the excellent application prospect of hepcidin as a substitute for antibiotics to resist pathogenic microorganisms in teleosts.

## 1. Introduction

In recent years, outbreaks of aquatic animal diseases caused by pathogenic microbial infections have posed a major problem for the aquaculture industry, resulting in enormous production and economic losses [1,2]. The usage of antibiotics is becoming more frequent; however, problems caused by this situation arrive one after another, such as antibiotic resistance [3], drug residues [4], damage to water ecosystems [5], and risks to human health through bioconcentration [6,7]. Therefore, there is an urgent need to find more effective treatments to substitute for antibiotics in the aquaculture industry. Antimicrobial peptides (AMPs) have the potential to act as substitutes for antibiotics because of their good biological properties [8,9], such as their small biomass, good stability, broad-spectrum antimicrobial potency, and low drug resistance, as well as their environmentally friendly characteristics [10,11,12]. Fish-derived AMPs are abundant. They may be safe and harmless to fish, and play a stronger role in the body’s immune regulation. Moreover, some species-specific peptides can promote fish cope with the complex water environment and microorganisms. In addition, the antibacterial spectrums of different AMPs vary with different species, and it is still necessary to strengthen the research of novel AMPs. The research on fish-derived AMPs not only enriches the resource pool of fish AMPs but also provides an important basis for differentiated application in fish healthy culture in the future.

A variety of AMPs are involved in the immune regulation of the organism against pathogenic bacterial infections in teleosts [13]. Among these AMPs, hepcidin performs an irreplaceable role in the innate immune defense system [14,15]. Human *hepcidin-25* (Hepc-25), as a single copy gene, has a dual function: regulating iron homeostasis and antimicrobial activity [16,17]. However, due to the degradation of the Hepc-25 maturation peptide, three isoforms of it, Hepc-20, Hepc-22, and Hepc-25, can be identified in plasma and urine [18,19]. Unlike other mammals, two different *hepcidin* genes (*Hecp1* and *Hecp2*) exist in mice that have different homologies with Hepc-25 [20,21]. Variably, two or more homologous *hepcidin* genes have been identified in teleosts, which are divided into two groups: HAMP1-type isoforms and HAMP2-type isoforms [22,23]. As described in previous studies, the HAMP1-type isoformscontains an N-terminal sequence (Q-S/I-H-L/I-S/A-L) that is similar to the DTHFP motif [24,25]. In mammals, the DTHFP motif is essential to the interaction of hepcidin with the sole cellular iron exporter ferroportin 1 (FPN1) in order to be involved in iron regulation [26,27,28], whereas, the HAMP2-type isoforms, with multiple copies, performs a unique role in the immune response against various pathogens [29,30,31]. In teleosts, Hepcidin not only performs antimicrobial function, including anti-bacterial [25,32], anti-viral [33,34,35], anti-fungal [36,37], and anti-parasitic activities [38], but it also possesses an immunomodulatory ability [36,39]. Up to now, numerous hepcidins have been identified in teleosts, such as zebrafish (*Danio rerio*) [40], tilapia (*Oreochromis mossambicus*) [41,42], red sea bream (*Pagrus major*) [43], European seabass (*Dicentrarchus labrax*) [44,45], half-smooth tongue sole (*Cynoglossus semilaevis*) [46], large yellow croaker (*Larimichthys croceus*) [47], orange-spotted grouper (*Epinephelus coioides*) [34,48], and spotted scat (*Scatophagus argus*) [32]. Therefore, Hepcidin is a promising alternative to antibiotics in light of its pivotal role in the antimicrobial and immune response. However, its mechanisms in disease resistance and immunity in fish remain ill-defined.

Golden pompano (*Trachinotus ovatus*), which belongs to the Carangidae family and Perciformes order, is regarded as an important mariculture economic fish in coastal areas of South China [49,50,51]. With the expansion of the culture scale, the outbreak of bacterial diseases has caused substantial production declines and resulted in huge financial losses [52]. Similar to other teleosts, there is an urgent need to investigate the immune defense mechanisms of *T. ovatus* and, in turn, provide theoretical support for disease prevention and treatment in *T. ovatus*. In this study, a derived antibacterial peptide of *T. ovatus* hepcidin2, TroHepc2-22, was chemically synthesized, and the biological functions were investigated through in vivo and in vitro assays. Our study elucidates the mechanisms of synthetic peptides in the defense against important pathogenic bacteria of marine fish, and its results will be valuable as a reference for future applications of hepcidin in fish.

## 2. Results

### 2.1. Peptides Synthesis and Structure Analysis

TroHepc2-22 is a cationic polypeptide with 22 amino acid residues, and it has the biochemical properties of a small molecular weight, high stability, and strong amphiphilicity (Table 1). The content of cysteine (Cys) in its amino acid composition is the highest, accounting for 36.4%. The projection of the helical wheel showed that TroHepc2-22 has an amphiphilic structure, as hydrophobic and hydrophilic residues were located on both sides of the wheel (Appendix A). Furthermore, visualized by Pymol-2.5.2 software, TroHepc2-22 contains a single hairpin structure formed by four disulfide bonds that consists of eight cysteine residues through two β-folds and a loop structure (Appendix A).

### 2.2. Antibacterial Activity of TroHepc2-22

The manifested inhibitory zone for six examined bacteria (*Escherichia coli* ATCC8739; *Edwardsiella piscicida*; *Vibrio harveyi* QT520; *Vibrio alginolyticus* HN08155; *Streptococcus agalactiae* LFY-5; *Staphylococcus aureus* ATCC6538) appeared on filter paper containing antibiotics (ampicillin or kanamycin, (positive control)) and TroHepc2-22; however, there was no inhibitory phenomenon around the P86P15 (Figure 1A), Hepc-25 (Figure 1B), and PBS. The results of the inhibitory zone assay indicated that TroHepc2-22 had a high antimicrobial activity on four tested bacteria, which included two Gram-positive (*S. agalactiae* and *S. aureus*) and two Gram-negative (*E. piscicida* and *V. harveyi*) bacteria (Figure 1).

To further examine the antimicrobial spectrum of TroHepc2-22, we evaluated and determined the minimum inhibitory concentration (MIC) and minimum bactericidal concentration (MBC) assays against the six tested bacteria, as described above. The result showed that TroHepc2-22 exerted varying degrees of inhibition effects against the tested bacteria (Table 2). The MIC and MBC values showed some variation, ranging from 8 to 128 μM and from 16 to 256 μM, respectively. As expected, the minimum values of the MIC (8 μM) and MBC (16 μM) were observed for *S. aureus*; however, the maximum values of the MIC (128 μM) and MBC (256 μM) were observed for *E. coli*, which was in line with those obtained from the inhibitory zone assay. For the Gram-negative bacteria, the MIC values ranged from 16 to 256 μM, and the values of the MBC were from 64 to 256 μM. In contrast, the MIC values were within the range of from 8 to 16 μM and the MBC values were from 16 to 32 μM for the Gram-positive bacteria. However, the MIC and MBC values of Hepc-25 were showed a higher degree. Except for the MIC value of *V. harveyi* was 128 μM, the MIC values of other bacteria were more than 256 μM (Table 1). Meanwhile, the values of the MBC of all four tested bacteria were more than 256 μM, which was in line with those obtained from the inhibitory zone assay. These results suggest that TroHepc2-22 possess a superior antibacterial function, while it was found to have better inhibitory and bactericidal effects against Gram-positive bacteria compared with Gram-negative bacteria.

### 2.3. Effects of TroHepc2-22 on Bacterial Growth

A bacterial growth inhibition test was performed to further verify the antibacterial activities of the synthetic polypeptide TroHepc2-22 against *S. aureus*, *S. agalactiae*, *E. piscicida*, and *V. harveyi*. The results showed that the growth of the four bacteria was completely inhibited when incubating with 1 × MIC TroHepc2-22 within 12 h, as shown in Figure 2. Moreover, TroHepc2-22 inhibited the bacterial growth in different degrees when the concentration was reduced to 1/4 × MIC and 1/2 × MIC. However, it still exhibited an inhibitory effect on the bacterial growth rate. Likewise, the effects of TroHepc2-22 on the bacterial growth exhibited a dose-dependent manner. Furthermore, compared with the Hepc-25, P86P15, and PBS group, TroHepc2-22 still had a significant inhibitory influence on the growth of several tested bacteria at a 1/4 × MIC concentration. However, no matter if it was P86P15 or Hepc-25, they could not inhibit the normal bacterial growth (Figure 2).

### 2.4. Bacterial Membrane Depolarization Induced by TroHepc2-22

Impacts on the depolarization of the cytoplasmic membrane were monitored using DiSC_3_(5), a probe molecule that detects and measures changes in the mitochondrial transmembrane potential, which were induced by changes in cell membranes [53]. We determined the membrane depolarization levels of *S. aureus*, *S. agalactiae*, *E. piscicida*, and *V. harveyi* incubated with TroHepc2-22 via fluorescence spectroscopy in the presence of DiSC_3_(5). The results revealed that the fluorescence intensity increased dramatically after adding different concentrations of TroHepc2-22 compared with the synthetic polypeptide P86P15 and PBS group (Figure 3). Meanwhile, the fluorescence intensity was gradually enhanced accompanied by the increased TroHepc2-22 concentration, indicating that the influence of TroHepc2-22 on the membrane depolarization was carried out in a certain dose-dependent manner. Apart from that, the depolarization of the Gram-positive bacteria was more intense than that of the Gram-negative bacteria under corresponding TroHepc2-22 concentration levels. Notably, the TroHepc2-22-induced level of membrane depolarization was the mildest for *E. piscicida* among the four tested bacteria.

### 2.5. PI Staining Analysis

To further verify the damage to bacterial membranes, the bacteria were incubated with TroHepc2-22 and then stained with propidium iodide (PI) dye. A fluorescence microscope observation showed that the amounts of PI-stained bacteria increased substantially after the TroHepc2-22 treatment, compared with the P86P15 and PBS groups (Figure 4).

### 2.6. Scanning Electron Microscopy (SEM) Observation

The surface ultra-structures of the bacteria incubated with TroHepc2-22 were monitored by scanning electron microscopy (SEM) examination. The results were shown in Figure 5. In the P86P15 or PBS group, the surfaces of the bacteria maintained a smooth and satiated state, and both the shapes and cell membranes of the bacteria remained fairly intact. In contrast, bacteria that were treated with 4 × MIC TroHepc2-22 had surface structural damages. More specifically, these bacterial cells presented irregular changes in morphology and visualized structural alterations. For example, in *S. aureus* and *S. agalactiae* (Figure 5C,F) treated with TroHepc2-22, their cells shrank, and their cell contents leaked out. As for the *E. piscicida* cells (Figure 5I), their cell surfaces formed vesicular protrusions, and the leakage of the cellular contents appeared. In addition, as revealed in Figure 5L, cell lysis and structure collapse emerged in *V. harveyi* when treated with TroHepc2-22. Therefore, we inferred that TroHepc2-22 could alter the morphological structure of the bacteria we tested, causing bacterial membrane rupturing and the leakage of the cytoplasm.

### 2.7. Binding Activity of TroHepc2-22 on Bacterial Genomic DNA

Based on the above experimental results, TroHepc2-22 had a relatively high capacity to disrupt the cell membrane integrity of the tested bacteria. We used a gel retardation method to detect whether the synthetic peptide could bind to bacterial genomic DNA and subsequently cause its degradation after penetrating the bacterial membrane. With regard to *S. aureus*, *S. agalactiae*, and *V. harveyi*, their gDNA was completely or partially degraded when treated with TroHepc2-22 at concentrations of ≥16 μM or ≥2 μM, respectively, while it showed little if any hydrolytic activity at concentrations of ≤2 μM (Figure 6A,B,D). However, at concentrations of ≥32 μM, 8–16 μM, and ≤4 μM, TroHepc2-22 caused the complete degradation of gDNA, and partial or no blockage of DNA migration for *E. piscicida* (Figure 6C), respectively. In terms of the control groups, DNAase I and P86P15 (Figure 6) caused the complete degradation of the gDNA or no blockage of the gDNA migration of the four tested bacteria, respectively.

### 2.8. Effects of TroHepc2-22 on Bacterial Infection

To investigate the in vivo effects of TroHepc2-22, we documented the bacterial loads in the immune-related tissues, including the liver, spleen, and head kidney, of fish infected with *V. harveyi* at various time points. The results showed that at 9 h post-infection (hpi) and 12 hpi, the bacterial loads in all the tested tissues were significantly lower than those in the P86P15 and PBS groups. Furthermore, in the immune-related tissues, including the liver, spleen and head kidney, the bacterial loads in the TroHepc2-22-injected group decreased by approximately 3.50, 3.60, and 3.98 times at 9 hpi, respectively. At 12 hpi, the bacterial loads decreased by approximately 3.61, 4.06, and 4.55 times, respectively (Figure 7). However, no significant difference was shown in the bacterial loads between the P86P15-treated and PBS groups.

### 2.9. Trohepc2-22 Regulates Immune-Related Gene Expressions against V. harveyi Infection

Quantitative real time PCR (qRT-PCR) was applied to determine the immune-related gene expressions in fish tissues under treatment with TroHepc2-22 before *V. harveyi* infection. As we predicted, the qRT-PCR results showed that TroHepc2-22 significantly up-regulated the expressions of inflammatory cytokines, including tumor necrosis factor-α (*TNF-α*), interleukin 1-β (*IL-1β*), *IL*-*6*, and interferon-γ (*IFN-γ*). Similarly, the expression levels of the Toll-like receptor (TLR) pathway genes, including *TLR1* and myeloid differentiation factor 88 (*MyD88*), were significantly elevated compared with those of the P86P15 and PBS groups (Figure 8A,B). Furthermore, in comparison with the P86P15 treated and PBS groups, the expression levels of the immune-related genes changed significantly after the TroHepc2-22 injection, indicating that TroHepc2-22 may exert its antimicrobial capacity by regulating inflammatory cytokines and activating immune-related signaling pathways.

## 3. Discussion

Hepcidin, which is rich in positively charged amino acids, belongs to the cationic AMPs [54,55]. In mammals, *hepcidin* is a single copy gene that exhibits a dual function as an AMP and a regulator of iron metabolism [16,27]. In teleosts, HAMP type-2 hepcidin was reported to perform an antimicrobial function and to play a crucial role in innate immune responses [30,55,56,57,58]. In the current study, we synthesized and examined the antimicrobial activities of TroHepc2-22 (GIKCRFCCGCCIPRVCGLCCRF), 22 amino acid residues from a *T. ovatus* hepcidin2 mature peptide. Most hepcidin mature peptides contain eight cysteine residues that form four disulfide bonds (Cys1–Cys8; Cys3–Cys6; Cys2–Cys4; Cys5–Cys7), with highly conserved positions in different organisms [19,59]. Similar to other teleosts, the *T. ovatus* hepcidin2 mature peptide that contains 22 amino acid residues also has eight cysteine residues, which form disulfide bonds to enhance the structural stability and reduce protein degradation [59,60]. The hairpin β-sheet type structure that is formed with the disulfide bonds then considerably increases their antimicrobial properties in hepcidin [19,61]. Furthermore, various hepcidin AMPs form an amphiphilic structure when interacting with the target membrane [56,62]. To conclude, the physical and chemical properties of TroHepc2-22 are closely related to its biological activity during pathogenic microorganism infection.

Previous studies have shown that synthetic hepcidin peptides have antibacterial activities against different pathogens. In orange-spotted grouper, the synthetic peptide EC-hepcidin exhibited strong antibacterial activities against *V. vulnificus* and *S. aureus* [34]. It has also been reported that in European seabass, a derived peptide Hep1 was capable of protecting fish against *V. anguillarum* [62]. In large yellow croaker, PC-Hepc, the synthesized hepcidin peptides, showed strong antibacterial activities against some principal fish pathogens, including *Aeromonas hydrophila*, *V. alginolyticus*, *V. harveyi*, and *V. parahaemolyticus* [37]. Moreover, two kinds of synthesized peptides, SmHep1P and SmHep2P, which were derived from turbot hepcidin, also possessed antibacterial potency against *Micrococcus luteus*, *S. aureus*, *E. tarda*, and *V. anguillarum*. The effects were stronger against Gram-positive bacteria than Gram-negative bacteria [35]. Consistent with these findings, in the current study, an inhibitory zone assay showed that the synthetic peptide TroHepc2-22 exhibited stronger antibacterial activity against *S. aureus*, *Streptococcus agalactiae*, *E. piscicida*, and *V. harveyi*, compared with that against *V. alginolyticus* and *E. coli*. Meanwhile, our results indicated that Hepc-25 derived from human hepcidin did not inhibit four fish pathogens used in this study, and TroHepc2-22 derived from golden pompano hepcidin exhibited stronger antibacterial activity against fish pathogens, similar to previous studies, suggesting that the antibacterial spectrums of AMPs from different species do differ. Moreover, TroHepc2-22 showed more potent antibacterial activity against Gram-positive bacteria than Gram-negative bacteria, according to the MIC and MBC analysis. These results proved that synthetic hepcidin from various species sources exhibit target/substrate preference.

As previously reported, the effectiveness of the AMP function hinges upon diverse antimicrobial mechanisms [8,63,64], among which antimicrobial peptides exert their functions mainly through direct killing and immunomodulatory mechanisms [65,66], while hepcidin interacts with the cell membranes of pathogenic microorganisms through electrostatic interaction and subsequently kills them by disrupting their cell membrane structure in a membrane-permeable manner [63]. As reported, Hep1, a derived peptide from European seabass hepcidin, could alter the bacterial membrane permeability of *V. anguillarum* [62]. As cationic hepcidin peptides of turbot, SmHep1P and SmHep2P exhibited a binding ability to the membrane of *E. tarda* [35]. In our study, we found that TroHepc2-22 could induce bacterial membrane depolarization and damage the membrane integrity of the bacteria. Furthermore, by virtue of SEM observation, it was clear that the outer membrane of the tested bacterial cells treated with TroHepc2-22 were wrinkled and broken, bringing about the leakage of the cell contents and the collapse of overall the structures of the cells. Similar to our research, a derivative BtHepc that was derived from brown trout (*Salmo trutta*) hepcidin was reported to have the ability to disrupt the integrity of the bacterial outer membrane of *A. salmonicida* and *A. hydrophila* [67]. Moreover, it was reported that two derived peptides in turbot, SmHep1P and SmHep2P were capable of altering the surface structure and causing damage to the membrane of *E. tarda* [35]. All the results suggest that TroHepc2-22 performs antibacterial activities by inducing membrane depolarization, changing the bacterial membrane permeability, and rupturing the morphological structure of both Gram-negative and Gram-positive bacteria.

Apart from the membrane-permeable mechanism that has been proven as the primary effector mechanism of AMPs, several related mechanisms have been advanced and have triggered heated discussions, including inhibiting the synthesis of intra-cellular and extracellular biopolymers and affecting intracellular functions [68,69]. Hepc25, a synthetic peptide of humans, performed considerable antibacterial activity by efficiently binding to the DNA of several bacterial species, including *Bacillus subtilis*, *B. megaterium*, and *M. luteus* [17]. Likewise, in mudskipper, BpHep-1 and BpHep-2 gave rise to the degradation of the gDNA of *E. tarda*. Moreover, Hep25 and Hep20, two derivatives of barbel steed hepcidin were reported to have hydrolase activity of the gDNA of *A. hydrophila* [25]. Similar to these findings, our results unveiled that TroHepc2-22 had the ability to hydrolyze the gDNA of the tested bacteria in a dose-dependent manner. Based on these results, TroHepc2-22 may exert its antibacterial activity by disrupting the membrane structure of bacteria and, in turn, hydrolyzing the bacterial gDNA.

Apparently, cell membranes are not the only target of antimicrobial peptides. AMPs also impede the critical physiological metabolism of pathogens by acting on intracellular macromolecules such as nucleic acids and proteins, ultimately leading to bacterial death. In European seabass, the mortality rate of the fish injected with the synthetic Hep1 peptide decreased from 72.5% to 23.5% after vibriosis infection compared with the control group, indicating that the synthetic peptide could protect European seabass against *V. anguillarum* infection [62]. In turbot, SmHep1P and SmHep2P significantly reduced the amounts of *E. tarda* infected kidney, spleen, and liver compared with the control peptide-injected fish [35]. Similarly, the loads of *Flavobacterium columnare* in grass carp administered with CiHep protein were significantly lower than those of the control fish [70]. Additionally, injection with the Hamp2 peptide could alleviate *Photobacterium damselae* infection in European seabass [71]. In line with these observations, our results showed that the bacterial loads in the liver, spleen, and head kidney of fish administered TroHepc2-22 significantly decreased after *V. harveyi* infection. Evidently, TroHepc2-22 exerted important antibacterial activity against *V. harveyi* infection in vivo.

In addition to killing pathogens directly, AMPs can also exert their effects through the immunological regulatory mechanism [72,73]. Previous research describes that hepcidin could induce some immune-related gene expressions [41,70,74]. For instance, grass carp (*Ctenopharyngodon idella*) hepcidin protected fish against *Flexibacter calumnsris* infection and induced immune-related genes, including *IL-1β*, *TNF-α*, and *IL-8* [70]. In zebrafish genetically modified with TH1-5 (tilapia hepcidin1-5), the expressions of some immune-related genes, such as *IL-10*, *IL-22*, *IL-26*, *MyD88*, *TLR1*, *TLR3*, *TLR4*, *NF-κB*, *TNF-α*, and *lysozyme*, were higher than those in the control group [74]. In line with these results, our study showed that after being infected with *V. harveyi*, the *T. ovatus* injected with TroHepc2-22, which expressed some inflammatory chemokines (*IL-6*, *TNF-α, IL-1β*, and *IFN-γ*) and immune pathway related factors (*TLR1* and *MyD88*), was upregulated. *IL-6*, *TNF-α, IL-1β*, and *IFN-γ*, as pro-inflammatory mediators, play major roles in immune responses and possess a wide range of biological functions [75,76]. Toll-like receptors (TLRs) play an extremely important role in the innate immune response by sensing, recognizing and binding pathogen-related molecular patterns, and then transmitting relevant signals to the downstream immune cascade [77,78]. It has been reported that hepcidin transgenic animals can enhance their resistance to bacterial pathogens by regulating the immune genes (such as interleukins, *TNF-α* and *TLR*s) [74,79,80]. The transgenic TH2-3 (tilapia hepcidin2-3) zebrafish significantly up-regulated the expressions of *Myd88*, *TLR1*, and *TLR3* to resist *V. vulnificus* infection [80]. Chinese black sleeper (*Bostrychus sinensis*) injected with a synthetic BsHep peptide increased the activation levels of immune-related genes (*TLR1*, *TLR2*, *TLR5* and *MyD88*) in the TLR signaling pathway against *V. parahemolyticus* infection [81]. In this study, we found that the expressions of *IL-6*, *TNF-α*, *IL-1β*, *IFN-γ*, *TLR1*, and *MyD88* were increased after the TroHepc2-22 injection, suggesting that the synthetic peptide TroHepc2-22 could regulate the expressions of immune-related genes and, hence, participate in the antibacterial immune response of the organism. These results revealed that the TroHepc2-22 peptide could provide protection against *V. harveyi* infection in *T. ovatus* and improve the fish’s antibacterial capacity. Our study revealed that the synthetic peptide TroHepc2-22, as a cationic AMP, possessed the antimicrobial potency against Gram-positive and Gram-negative bacteria in vitro assays. Furthermore, in vivo TroHepc2-22 exerts a pivotal role in innate immunity by both protecting against *V. harveyi* infection and regulating the expression of immune-related genes. Hence, it is viable that TroHepc2-22 acted as a novel and promising candidate for the substitute for antibiotics.

## 4. Materials and Methods

### 4.1. Fish, Bacterial Strains and Culture Conditions

Healthy golden pompano (*T. ovatus*), weighing 15.3–16.5 g, were acquired from a commercial fish farm in Chengmai County, Hainan Province, China. The fish were reared at 26 °C in a filtered-seawater recirculating system for one week and acclimatized to the experimental conditions. Prior to tissue collection, tricaine methanesulfonate (Sigma, St. Louis, MO, USA) was applied for the euthanasia of the fish [48]. All experiments were conducted in compliance with the guidelines and regulations approved by the Animal Care and Use Committee of Hainan University (No. HNU200521).

Six bacteria were examined in this experiment, including four Gram-negative bacteria (*E. coli* ATCC8739; *E. piscicida*; *V. harveyi* QT520; and *V. alginolyticus* HN08155) and two Gram-positive bacteria (*S. agalactiae* LFY-5 and *S. aureus* ATCC6538). Among the six strains, *S. agalactiae*, *V. harveyi*, *V. alginolyticus*, and *E. piscicida* were cultured at 30 °C, while *S. aureus* and *E. coli* were cultured at 37 °C. *S. agalactiae* was cultured in a brain heart infusion (BHI) medium, and the remaining five bacteria were raised in Luria–Bertani (LB) medium.

### 4.2. Peptides Synthesis and Structure Analysis

The peptide TroHepc2-22 (GIKCRFCCGCCIPRVCGLCCRF) consists of 22 amino acid residues, which corresponds to the mature peptide of *T. ovatus* hepcidin2 (GenBank accession No.OM643385). TroHepc2-22, an N-terminal acetylated and C-terminal amidated linear peptide, was synthesized by GL Biochem (Shanghai, China). The peptide was purified to 96.66% via high performance liquid chromatography (HPLC), and it was identified using mass spectrometry (MS) analysis. The control peptide P86P15 (FKFLDNMAKVAPTEC), which is derived from a viral protein and is usually used as a negative control peptide, was synthesized in a similar manner [35,82,83]. Meanwhile, we synthesized the classical antimicrobial peptide Human hepcidin-25 (Hepc-25) in our studies. Based on previous research, Hepc-25 (DTHFPICIFCCGCCHRSKCGMCCKT) was synthesized by GL Biochem (Shanghai, China), with N-terminal acetylation and C-terminal modification [15,17,19]. We divided the synthesized peptides TroHepc2-22, Hepc-25, and P86P15 into smaller sample sizes and stored them in sealed containers containing desiccant at −80 °C. The dissolution of the synthesized peptides was performed using the method reported by Barroso [71], with small changes. Briefly, the peptides were taken in batches and dissolved in a sterile PBS solution. The peptides were sonicated at a low temperature to promote peptide solubilization and were finally filtered through a 0.2 µM pore size filter membrane to remove bacteria. ProtParam website (http://us.expasy.org/tools/protparam.htmL) (accessed on 21 April 2022) was used to analyze the molecular weight, isoelectric point, molecular formula, total atomic number, lipid coefficient, instability index, hydrophilicity index, and other related biochemical characteristics of TroHepc2-22 peptide. The properties such as hydrophobicity, hydrophobic moment, net charge, amino acid composition, and helical wheels of the peptide TroHepc2-22 were predicted and analyzed by HeliQuest online database (http://heliquest.ipmc.cnrs.fr/) (accessed on 2 May 2022). The three-dimensional structure was predicted using the PEP-FOLD3 server, and it was subsequently visualized using Pymol-2.5.2 software.

### 4.3. Antibacterial Activity of TroHepc2-22

#### 4.3.1. Inhibitory Zone Assay

A zone of inhibition assay was performed to determine whether the synthetic peptide TroHepc2-22 could inhibit bacterial growth, based on our previously conducted study [84]. In brief, the bacterial cells were rinsed and resuspended three times with a phosphate buffer saline (PBS) solution (Solarbio, Beijing, China), and the concentration of the suspension was subsequently diluted to 1 × 10^7^ colony-forming unit (CFU)/mL. Afterwards, 100 µL of the suspension of bacteria was applied evenly to the solid plate, followed by the affixation of the blank filter paper to the plates. A total of 20 µL of 1 mg/mL TroHepc2-22, Hepc-25, and P86P15 (negative control) or 1 mg/mL antibiotics (ampicillin or kanamycin) or PBS (blank control) were added dropwise onto the blank filter paper. Finally, the plates were incubated at 30 °C or 37 °C for 16 h, according to previous studies [84,85]. Subsequently, the bacteriostatic circles that appeared on the plates were successfully observed.

#### 4.3.2. Minimum Inhibitory Concentration (MIC) and Minimum Bactericidal Concentration (MBC) Assay

The MIC and MBC of TroHepc2-22 were measured in 96-well culture plates with a two-fold microdilution assay, based on previous description [86]. Briefly, the tested bacteria were cultured to an optical density of 0.6 at a wavelength of 600 nm (OD600) and subsequently resuspended to 2 × 10^6^ CFU/mL with PBS. Peptides TroHepc2-22 and P86P15 were prepared and diluted serially at two-fold from 256 μM to 1 μM. Afterward, 100 µL of the suspension of the tested bacteria was mixed with the same volume of the peptides TroHepc2-22 or P86P15 (negative control), or PBS solution (blank control), in 96-well microplates. After incubation, the absorbance of each well was observed visually, and then the OD600 values of each well was measured by a multi-wavelength microplate reader (BioTek, Winooski, VT, USA) at 600 nm. The MIC was determined as the minimal peptide concentration at which the complete inhibition of bacterial growth was observed. A mixture of 100 µL at or above the MIC value was taken and applied to solid plates and subsequently incubated for 16 h. The minimum concentration at which no bacterial growth was observed on the plates was defined as the MBC. The experiments were independently conducted and repeated thrice.

#### 4.3.3. Growth Curve Assay

The growth inhibition kinetics of the peptides on the bacteria were investigated as previously described [84]. Briefly, 100 µL of TroHepc2-22 was added to the same volume of bacterial (*S. aureus*, *S. agalactiae*, *E. piscicida*, and *V. harveyi*) suspension to obtain the final concentrations of 1/4 × MIC, 1/2 × MIC, and 1 × MIC in 96-well plates, and then the 96-well plates were cultured at 30 °C or 37 °C. Sterile PBS acted as a blank control. Hepc-25 and P86P15 with the same final concentration as the 1 × MIC of TroHepc2-22 were added to the tested bacteria as negative controls. Afterwards, the OD_600_ values of each well were determined by a multi-wavelength microplate reader for 12 consecutive hours at 1-h intervals. The experiment was independently conducted three times, each performed in triplicate, as previously observed [84,87]. This assay was replicated three times.

### 4.4. Antibacterial Mechanisms of TroHepc2-22

#### 4.4.1. Bacterial Membrane Depolarization Assay

Cytoplasmic membrane depolarization usually occurs because of the current flow outside the membrane or a change in the ionic composition of the external fluid. In this assay, 3,3′-dipropylthiadicarbocyanine iodide (DiSC_3_(5)) (Sigma, St. Louis, MO, USA), a membrane potential-sensitive cyanine dye, was used to assess whether the peptide had an effect on the bacterial membrane depolarization. Some modifications were performed on the assay, according to previous studies [88,89].

The tested bacteria in this experiment were washed thrice with buffer A (5 mM HEPES, 20 mM glucose) and then the Gram-positive bacteria were resuspended in buffer B (5 mM HEPES, 20 mM glucose and 100 mM KCl), and the Gram-negative bacteria were resuspended in buffer C (5 mM HEPES, 20 mM glucose, 100 mM KCl and 2 mM EDTA). Subsequently, 100 µL of the tested bacterial suspension (5 × 10^6^ CFU/mL), diluted as described above, was separately mixed with 50 µL of 1 μM DiSC_3_(5) stain and incubated for 90 min at room temperature. Continuous measurements of the fluorescence intensity were performed for 10 min at excitation and emission wavelengths of 622 nm and 670 nm, respectively. Subsequently, 50 µL of TroHepc2-22 was added to the 96-well plate to achieve final concentrations of 1/2 × MIC, 1 × MIC, 2 × MIC, and 4 × MIC, and then the change in this fluorescence value was monitored for 30 min. 50 µL of P86P15 with the same concentration of the corresponding bacterial 4 × MIC and PBS were then used as the negative and blank controls, respectively.

#### 4.4.2. Propidium Iodide (PI) Staining Assay

Propidium iodide (PI) is a nuclear staining reagent that indicates damage to the cell membrane. 4′,6-diamidino-2-phenylindole (DAPI) is a fluorescent dye that binds strongly to DNA, and it is commonly used to stain live cells and damaged cells under fluorescent microscopy [90,91]. For the PI staining assay, four tested bacteria (*S. aureus*, *S. agalactiae*, *E. piscicida*, and *V. harveyi*) were pretreated as described above, and they were then mixed with 150 µL of TroHepc2-22 (at its final concentration of 1 × MIC). Subsequently, the mixture was stained with PI (Sigma, St. Louis, MO, USA) and DAPI (Solarbio, Beijing, China) at final concentrations of 3 ng/mL and 5 ng/mL, respectively. In order to remove the remaining unbound dye, the suspension was rinsed with PBS after the mixture was incubated for 1 h at room temperature. The bacteria were then visualized under a florescent inverted microscope (Leica, Wetzlar, Germany) [92]. Compared with TroHepc2-22, the P86P15 and PBS performed as the negative and blank controls, respectively.

#### 4.4.3. Scanning Electron Microscopy (SEM) Visualization

Scanning electron microscopy (SEM) is widely used in modern cell biology research because of its stereoscopic imaging and its ability to directly observe the fine structure of the cell surface. Based on previous studies [35,58], SEM was applied to investigate the effect of the peptide TroHepc2-22 on the alternating morphologies of the tested bacteria. The bacterial suspension (1 × 10^8^ CFU/mL) after washing with 0.9% NaCl was incubated with TroHepc2-22 at a final concentration of 4 × MIC for 1 h at room temperature. P86P15 with the same concentration of the corresponding bacterial 4 × MIC and PBS performed as the negative and blank controls, respectively. Afterwards, the tested bacteria were washed thrice with 0.9% NaCl, and they were then fixed in 2.5% glutaraldehyde in the dark for 4 h at 4 °C. Subsequently, the bacterial cells washed with 0.9% NaCl were separately dehydrated with different concentrations (30%, 50%, 70%, 90%, and 100%) of ethanol for 15 min. Furthermore, 10 µL of bacterial cell suspension was dropped on tin foil and then lyophilized in a vacuum freeze drier (ALPHA 2-4 LD plus, Christ, Germany) for more than 12 h. After lyophilization, the specimens were treated with gold spray and then observed and photographed under SEM (Verios G4 UC, Thermo Sicentifc, Waltham, MA, USA).

#### 4.4.4. Gel Retardation Assay

To investigate whether TroHepc2-22 has hydrolase activity on bacterial genomic DNA, the gel retardation assay was performed as previously reported [57]. For the in vitro assay, bacterial genomic DNA from the tested bacteria were extracted with the TaKaRa MiniBEST Bacteria Genomic DNA Extraction Kit Ver.3.0 (Takara Bio, Beijing, China), and then the NanoPhotometer (NanoDrop2000c, Thermo Sicentifc, USA) was used to measure its quality and concentration. Bacterial gDNA (100 ng) was incubated with TroHepc2-22 (final concentrations at two-fold dilution from 64 μM to 1 μM) in a total volume of 10 µL. Meanwhile, the same volumes of 64 μM P86P15 and Dnase I were applied as the negative and positive controls, respectively. Finally, the mixture was incubated for 1 h at room temperature and subjected to 1.5% agarose gel electrophoresis. Bacterial gDNA agarose gel electrophoresis maps were obtained by gel imager photography (Gel Doc XR, BIO-RAD, Hercules, CA, USA).

### 4.5. In Vivo Study on Pathogens Infection

To ascertain the in vivo effects of TroHepc2-22 on bacterial invasion, the pathogens infection test was conducted based on previously reported [47,70]. Briefly, 45 fish were randomly divided into three groups (A, B, and C). Group A was intraperitoneal (i.p.) injected with 100 µL of TroHepc2-22 (200 μg/mL), group B (negative control) was injected i.p. with 100 µL of P86P15 (200 μg/mL), and group C was injected with 100 µL of PBS as blank control. Fish of three groups were inoculated via i.p. injection with 100 µL of *V. harveyi* (1 × 10^6^ CFU/mL) at 12 h post-infection (hpi). Subsequently, the liver, spleen, and head kidney of 15 fish were collected at 6, 9, and 12 hpi. Five identical tissue samples from each group were aseptically mixed to a sample for each time point. At 12 hpi, liver, spleen, and head kidney were divided into two parts, one of which was used for bacterial loads detection and the other was stored in RNAfixer (Takara Bio, Beijing, China) for a following detection of immune-related gene expressions. The plate counting method was performed to examine bacterial loads in the tissues based on previously described [93].

### 4.6. Expression of Immune-Related Genes Induced by TroHepc2-22

To determine the impact of synthetic peptides on immune-related genes, the liver and head kidney were obtained as 4.5 described. Total RNA was extracted, and was then reversed to synthesize complementary DNA (cDNA), which was used as a template for qRT-PCR reactions [94]. The mRNA expression levels of immune-related genes, including *TNF-α*, *IL-1β*, *IL-6*, *IFN-γ*, *TLR1*, and *MyD88*, were examined by qRT-PCR using the 2^−ΔΔCt^ method. Meanwhile, the housekeeping gene was *β-2-microglobulin* (*B2M*), based on previous studies [95,96]. All primers used in this experiment were presented in Table 2.

### 4.7. Statistical Analysis

Statistical analysis was performed with the SPSS 16.0 program (Chicago, IL, USA) and GraphPad Prism 8.0.0 (San Diego, CA, USA). Data were analyzed using ANOVA. Statistical significance was evaluated with a *p* value less than 0.05.

## 5. Conclusions

Consistently, the findings of this study indicated that TroHepc2-22, as a cationic and strong stability AMP, exhibited antibacterial activities against Gram-positive and Gram-negative bacteria. On the one hand, TroHepc2-22 induced the depolarization of the bacterial membranes, altering the bacterial membrane permeability and modifying the morphological structure. On the other hand, TroHepc2-22 interacted with the bacterial genomic DNA and degraded it. Therefore, TroHepc2-22 exerts strong antibacterial activity and regulates the expressions of immune-related genes in in vitro and in vivo experimental analyses. In conclusion, we found a derived hepcidin peptide, TroHepc2-22, which has great potential as a substitute for antibiotics to help prevent antibiotic abuse.

## Figures and Tables

**Figure 1 ijms-24-09251-f001:**
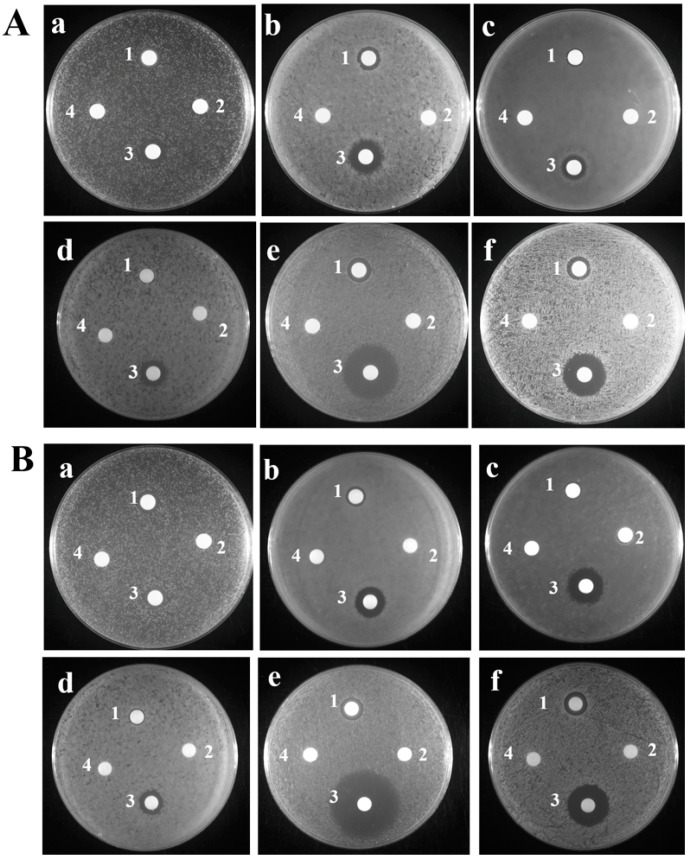
Inhibitory Zone Assay of TroHepc2-22. (**A**): (a) *Edwardsiella piscicida*; (b) *Vibrio harveyi*; (c) *V. alginolyticus*; (d) *Escherichia coli*; (e) *Streptococcus agalactiae*; (f) *Staphylococcus aureus*. (1) 1 mg/mL TroHepc2-22; (2) 1 mg/mL P86P15; (3) 1 mg/mL ampicillin for (a) *E. piscicida* and (e) *S. agalactiae*; and 1 mg/mL kanamycin for (b) *V. harveyi*, (c) *V. alginolyticus*, (d) *E. coli* and (f) *S. aureus*; (4) PBS. (**B**): (a) *E. piscicida*; (b) *V. harveyi*; (c) *V. alginolyticus*; (d) *E. coli*; (e) *S. agalactiae*; (f) *S. aureus*. (1) 1 mg/mL TroHepc2-22; (2) 1 mg/mL Hepc-25 (3) 1 mg/mL ampicillin for (a) *E. piscicida* and (e) *S. agalactiae*; and 1 mg/mL kanamycin for (b) *V. harveyi*, (c) *V. alginolyticus*, (d) *E. coli* and (f) *S. aureus*; (4) PBS.

**Figure 2 ijms-24-09251-f002:**
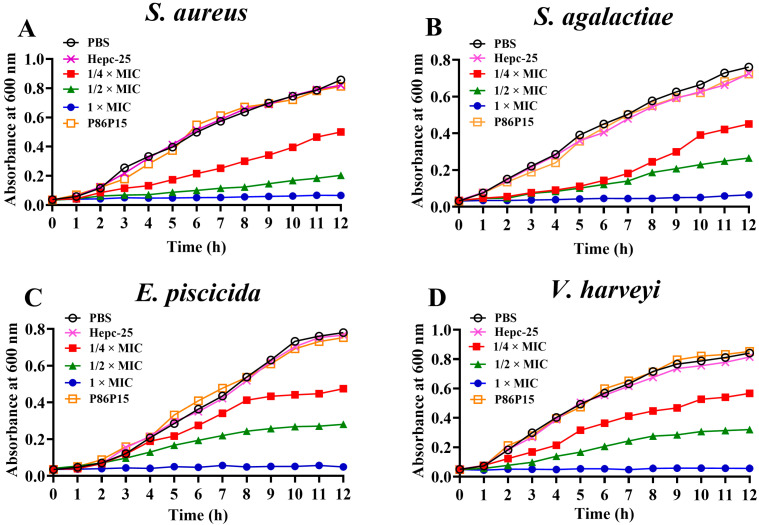
Growth curve of TroHepc2-22 against four tested bacteria. Four tested bacteria including (**A**) *S. aureus*; (**B**) *S. agalactiae*; (**C**) *E. piscicida*; and (**D**) *V. harveyi* were added to 100-well plates, bacteria were mixed with 100 μL TroHepc2-22 (final concentration at 1, 1/2, and 1/4 × MIC). With the corresponding concentration Hepc-25, P86P15 or PBS as negative control and blank control, respectively.

**Figure 3 ijms-24-09251-f003:**
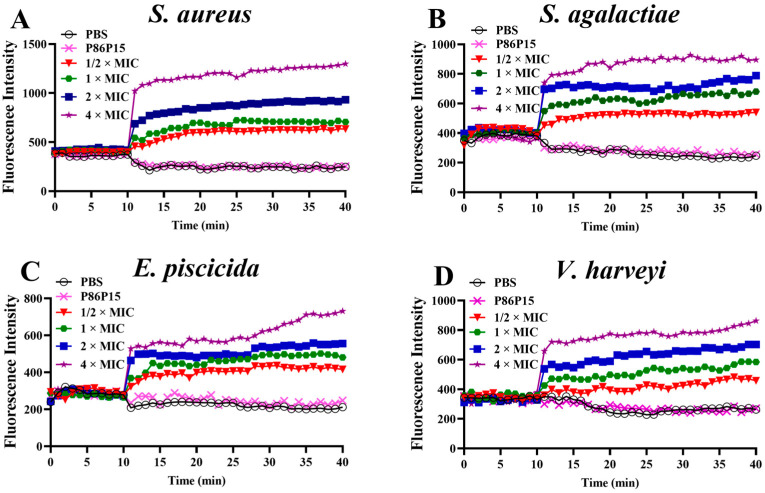
Cytoplasmic membrane depolarization of four tested bacteria examined by using the voltage-sensitive probe DiSC_3_(5). Four tested bacteria including *S. aureus*, *S. agalactiae*, *E. piscicida*, and *V. harveyi* were treated with TroHepc2-22 at 1/2, 1, 2, 4 × MIC in the 10th minute, with the addition of P86P15 at the corresponding concentration or PBS as negative control and blank control, respectively. (**A**) *S. aureus*; (**B**) *S. agalactiae*; (**C**) *E. piscicida*; and (**D**) *V. harveyi*.

**Figure 4 ijms-24-09251-f004:**
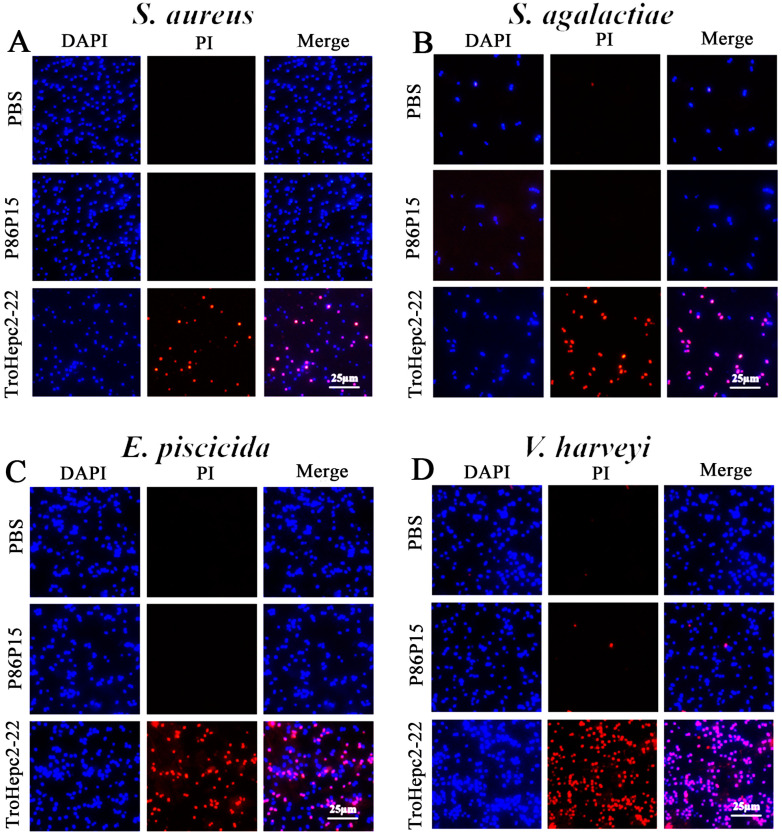
Propidium iodide staining analysis of bactericidal activity of TroHepc2-22. Four tested bactericidal, including (**A**) *S. aureus*; (**B**) *S. agalactiae*; (**C**) *E. piscicida*; and (**D**) *V. harveyi* with 1 × MIC TroHepc2-22 or the same concentration of P86P15 (negative control) or PBS (blank control) for 1 h, respectively. Following the 1 h incubation, bacteria were stained with DAPI and propidium iodide and observed under a fluorescence microscope.

**Figure 5 ijms-24-09251-f005:**
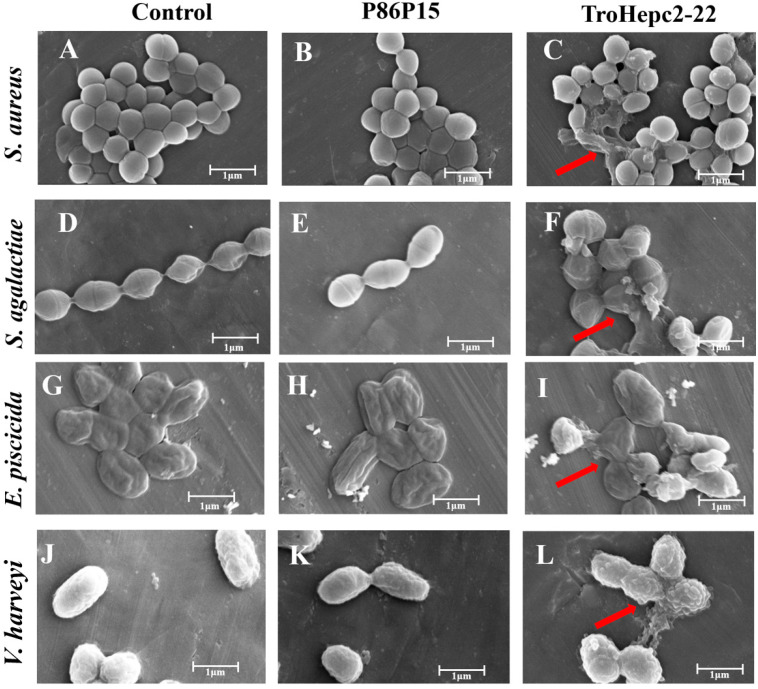
The observation on four bacterial cells with or without the treatment of TroHepc2-22 (4 × MIC) was based on SEM at the magnification of 25,000×. Four tested bacteria including *S. aureus* (**A**–**C**), *S. agalactiae* (**D**–**F**), *V. harveyi* (**G**–**I**), and *E. piscicida* (**J**–**L**) were treated with TroHepc2-22 at a concentration of 4 × MIC or 50 μL of P86P15 with the same concentration of corresponding bacterial 4 × MIC and PBS were used as negative control and blank control, respectively. TroHepc2-22 alter the morphological structure of bacteria we tested, causing bacterial membrane rupturing and leakage of the cytoplasm. The red arrows indicate that *S. aureus* cells were shrunk and their contents leaked out (**C**), *S. agalactiae* cells morphological structure were altered, disrupted, and their contents leaked out (**F**), *E. piscicida* cells surface formed vesicular protrusions and cellular contents leaked out (**I**), and *V. harveyi* cells lysis and their structure collapse emerged (**L**).

**Figure 6 ijms-24-09251-f006:**
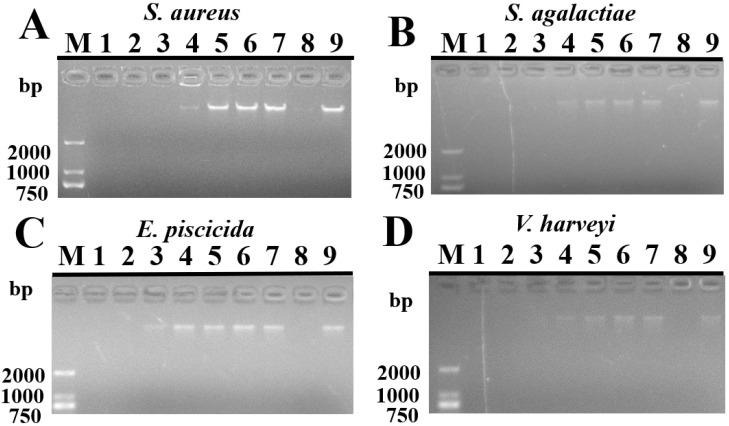
Effects of TroHepc2-22 on bacterial genomic DNA. Extracted genomic DNA of four tested bacteria were incubated with 1–64 μM TroHepc2-22 (lane 1–7: 64, 32, 16, 8, 4, 2, 1 μM, respectively), Dnase I (lane 8) and 64 μM of P86P15 (lane 9) for 30 min at room temperature. (**A**) *S. aureus*; (**B**) *S. agalactiae*; (**C**) *E. piscicida*; and (**D**) *V. harveyi*.

**Figure 7 ijms-24-09251-f007:**
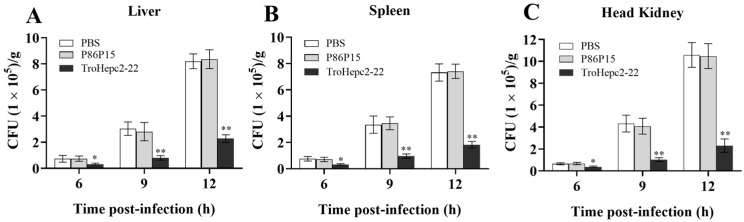
Impact of TroHepc2-22 on *V. harveyi* infection. *T. ovatus* were administered intraperitoneal injection with TroHepc2-22, P86P15 and PBS before being infected with *V. harveyi*. Bacteria loads in liver (**A**) spleen (**B**) and head kidney (**C**) of *T. ovatus* were counted at 6, 9, and 12 h post-infection. Data in this assay were expressed as mean ± SEM (N = 5). N, the number of fish was performed. The statistical significance was marked with “*”. (**) *p* < 0.01, (*) *p* < 0.05.

**Figure 8 ijms-24-09251-f008:**
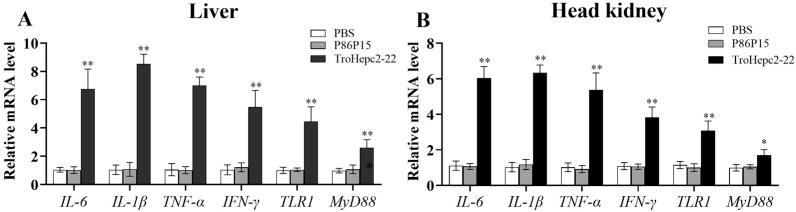
Expression level of immune-related genes of *T. ovatus* injected with TroHepc2-22. Sample infections with *V. harveyi* or PBS in liver (**A**) and head kidney (**B**) were determined, and were analyzed by qRT-PCR, with the mRNA level of the PBS-treated group as 1. Data in this assay were expressed as mean ± SEM (N = 5). N, the number of fish was performed. The statistical significance was marked with “*”. (**) *p* < 0.01, (*) *p* < 0.05.

**Table 1 ijms-24-09251-t001:** Physical and chemical parameters of derived peptides to TroHepc2-22.

Physical and Chemical Parameters	TroHepc2-22
Molecular formula	C_100_H_166_N_32_O_23_S_8_
Total atomic number	329
Net electric charge	+4
Amino acid residue	22
Molecular weight	2441.10
Constant electric point	8.78
Coefficient of fat	66.36
Coefficient of instability	−2.91
Hydrophilicity index	1.018
Hydrophobic index	0.869
Hydrophobic moment	0.244

**Table 2 ijms-24-09251-t002:** Antibacterial activity of the synthetic TroHepc2-22 and Hepc-25.

Microorganisms	Bacterium	TroHepc2-22 MIC (μM)	TroHepc2-22 MBC (μM)	Hepc-25 MIC (μM)	Hepc-25 MBC (μM)
Gram-positive bacteria	*Streptococcus agalactiae*	16	32	>256	>256
	*Staphylococcus aureus*	8	16	>256	>256
Gram-negative bacteria	*Edwardsiella piscicida*	32	64	>256	>256
	*Vibrio harveyi*	16	32	128	>256
	*Vibrio alginolyticus*	64	128	>256	>256
	*Escherichia coli*	128	256	>256	>256

## Data Availability

The authors declare that all data supporting the findings of this study is available within the article.

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
