# Peer review of "Antibacterial Activity and Mechanisms of TroHepc2-22, a Derived Peptide of Hepcidin2 from Golden Pompano (Trachinotus ovatus)"

_ijms, 2023, doi:10.3390/ijms24119251_

Round 1
Reviewer 2 Report
I have read the manuscript entitled “Antibacterial Activity and Mechanisms of a Derived Peptide Trohepc2-22 from Golden Pompano Hepcidin” with great interest and I think it is in principle suited for a publication in the IJMS. The authors have analyzed novel derived antimicrobial peptide against both Gram-negative and Gram-positive bacteria and demonstrated possible mechanism action of the AMP. The paper brings some new information to the scientific community. I am happy that figures, tables, and methods are well presented. I can suggest that manuscript can be published after minor text editing.
Minor Comments:
Line 60: “…contain a N-terminal sequence (Q-S/I-H-L/I-S/A-L)…” maybe you mean QSHLS? Please check.
Lines 65-66: “…anti-fungi [36-37]…” I think that in the cited article #36 antifungal effects are not demonstrated. Please check.
Reviewer 3 Report
This manuscript is aimed at benefitting the aqua culture industry, which affects a lot of communities. The authors have explained the hypothesis with adequate introduction, supported it with suitable methods and provided decent discussion and conclusion.
Here are my major concerns:
1. The authors selected a portion from the Hepcidin sequence of Golden Pompano, and not synthesized/designed a novel peptide sequence.
2. It is not clear whether that single peptide is linear or was cyclized/oxidized to make the disulfide bonds. If it was kept linear, was it deliberate and what is the rationale behind it? Because it is very difficult to prevent the peptides containing multiple cysteine residues from getting oxidized/cyclized in solution.
3. It is not clear whether they synthesized the peptide or procured it from commercial sources. If procured from outside, it has to be written accordingly.
4. What is the rationale behind selecting P86P15 as a control? This peptide has to be introduced with adequate references along with its amino acid sequence.
5. Most graphs have only one control and there is an overlap with P86P15 with PBS. My understanding is that PBS is the media and serves as a negative control. In any case, both positive and negative controls have to be added for all the graphs and the controls have to be clearly defined (and not just as “control” or “control group”).
Here are my other suggestions:
1. The title can be slightly modified to make it clearer.
2. The abstract has a lot of words that can be replaced to make it sound more logical. Sections 2.7, 2.8, and 2.9 can also be rewritten to make for better understanding. Overall, the English and grammar can be improved.
3. Title has to be added for Table 2.
4. Please correct P68P15 in Figure 3 to P86P15.
5. Please add clearer images for Figure 5 since the current ones are not very clear.
6. Title for section 4.4 is wrong and has to be fixed.
Round 2
Reviewer 1 Report
The authors responsed most of the concerns and sounds reasonable, but for the second question, the answer is not enough and a control peptide of hepcidin should be included to compare with.
Round 3
Reviewer 1 Report
A evolutionary comparison of hepcidin from different speices may be a interest point for future research, to unveiling the specie-specific reason.